# *Salvia mellifera*—How Does It Alleviate Chronic Pain?

**DOI:** 10.3390/medicines6010018

**Published:** 2019-01-24

**Authors:** James David Adams, Steven Guhr, Enrique Villaseñor

**Affiliations:** 1Department of Pharmacology and Pharmaceutical Sciences, School of Pharmacy, University of Southern California, Los Angeles, CA 90089, USA; guhr@usc.edu; 2Stevenson Ranch, CA 91381, USA; kayakerev@gmail.com

**Keywords:** *Salvia mellifera*, black sage, pain chemokine cycle, chronic pain

## Abstract

Black sage, *Salvia mellifera*, can be made into a sun tea that is used as a foot soak to treat pain patients. The monoterpenoids and diterpenoids in the preparation penetrate the skin of the feet and stop the pain chemokine cycle, which may be the basis of chronic pain. Several chronic pain patients have reported long-term improvements in their pain after treatment with the preparation.

## 1. Introduction

Black sage, *Salvia mellifera*, is a traditional medicine of the Chumash Indians of California [1,2]. It is used as a sun tea made from the stems and leaves of the plant to treat pain. *S. mellifera* sun tea is also traditionally used to cure chronic pain. The authors have used this and other Chumash medicines to treat many pain patients. Black sage contains 54 monoterpenoids and several diterpenoids such as carnosol (41%), carnosic acid (22%), salvicanol (15%) and rosmanol (9%) [3,4]. The monoterpenoids are 1.8-cineole (39.8%), camphor (12.2%), α-pinene (9.2%), limonene (2.2%), myrcene (2%), γ-terpinene (2%), terpene-4-ol (2%) and many less abundant monoterpenoids. 

Currently in the US, chronic pain affects 60%, or more, of people over the age of 65 [5]. Chronic pain is pain that continues long after the initial cause of the pain is gone, such as back injuries, car accidents, surgery, nerve damage and infections. Unfortunately, there is no cure for chronic pain—only partial short-term therapies are available—but chronic postsurgical pain can sometimes be prevented [6]. 

The causes of chronic pain are only now being understood and may involve chemokines. It has been proposed that peripheral and central sensitization mechanisms perpetuate chronic pain [6,7] but none of these mechanisms adequately explain how to cure chronic pain. Opioids and other drugs are used to manage pain and chronic pain [6]. Patients seek these drugs that cause 100,000 or more deaths every year [8]. In the US, there is an opioid crisis and a non-steroidal anti-inflammatory drugs (NSAIDs) crisis due to excessive use of these dangerous oral drugs. Many patients believe that pain comes from the brain. Therefore, pain must be treated with drugs that penetrate into the brain.

Pain is felt in the skin due to the abundance of pain receptors in the skin, such as transient receptor potential cation (TRP) channels, prostaglandin receptors, histamine receptors, muscarinic receptors and many more [9,10]. The safest and most effective treatment for pain is to apply a medicine to the skin [9,10]. Topical medicines are safer than oral medicines and could save the lives of thousands of pain patients. The dilemma with topical pain medicines is to find potent medicines that can treat even severe pain, yet do not have toxicity problems. Sagebrush liniment contains cineole, which is more powerful than morphine [11] and is used by topical application to treat broken bones, gunshot wounds, cancer pain and other severe pain [11,12]. Sagebrush liniment can also provide long-term relief from chronic pain. Several topical pain medicines are commercially available with more under development [13].

Chronic pain may be caused by a pain chemokine cycle that involves the release of chemokines in the skin by damaged or stressed cells [8,14]. Chemokines attract macrophages to the skin and induce cyclooxygenase-2 (COX-2) in macrophages, which release prostaglandins. Prostaglandins cause pain by binding to prostaglandin receptors and prolong pain by inducing the phosphorylation of TRP channels [11]. This activates TRP channels and makes them more sensitive to stimuli. Phosphorylation of Na^+^ channels is also induced by prostaglandins [15], which may make them more sensitive to stimuli. Both TRP and Na^+^ channels, as well as many other skin receptors, are important in pain [10]. Prostaglandins also enhance the release of chemokines in the skin. Chemokines cause the activation of TRP channels to increase and prolong pain. Macrophages secrete IL-23 and IL-1β that induce the synthesis of IL-17 by skin resident T cells. IL-17 induces chemokine release in the skin. Chemokines stimulate the release of IL-17. This establishes a self-perpetuating pain chemokine cycle in which prostaglandins, chemokines and IL-17 cause, enhance and prolong pain. The skin produces pain during chronic pain. Curing chronic pain involves inhibiting TRP channels, chemokine production, IL-17 production, COX-2 expression and perhaps other mechanisms [8,16].

The brain may be involved in the pain chemokine cycle (Figure 1). The activation of afferent sensory neurons in the skin leads to chemokine release in the brain [17]. Chemokines in the brain modulate the actions of other neurons, including descending and peripheral neurons, which may result in chemokine release in the skin [17]. These chemokines can be released in sites distant from the site of application of medicine to the skin.

## 2. The Medicine 

Black sage sun tea is made by collecting fresh branches and leaves of *S. mellifera*, black sage. This plant material, 115 g, is put into 2 L of fresh or sea water. This is put under the sun for 6–8 h to make a sun tea. The plant material is removed from the preparation prior to use. A patient soaks both feet in the sun tea for 20 min. The sun tea is then stored in a refrigerator until the next day. The patient uses the same sun tea as a foot soak for 20 min. This is repeated daily for a total of about seven treatments. Patients were also advised to perform 20 min or so of mild exercise daily, such as walking. All patients were asked to read and sign an informed consent form that guarantees the privacy of their information. The sagebrush liniment is described in other publications [11,12]. None of the preparations described in this work are commercially available. The name of the ethics committee: University of Southern California Institutional Review Board, ethical approval code: Application HS-19-00042.

## 3. Case Reports: Use of Black Sage Sun Tea in Pain Patients

There are reports of pain patient treatment with black sage sun tea and long-term improvements in chronic pain with the medicine [1,11,16]. The following are previously unreported patient results. Patients were recruited at talks about traditional medicine given by the authors to the public (Table 1). The authors are not medical doctors and did not perform any diagnostic tests or examinations on any patient. Patients reported the diagnoses they had been given by other medical professionals. Most patients declined to discuss their family histories, medications or other health conditions. The authors did not examine patients after treatment but relied on the ability of patients to rate their pain on a scale of 0–10. Many patients were treated one time with black sage sun tea, reported pain relief, but did not respond to requests for follow-up information. These patients are not included in the current report. The total number of patients recruited was probably between 100 and 200.

A 20-year-old Caucasian woman suffered from a sports injury that resulted in compartment syndrome in both legs. She had surgery to relieve the edema in her legs. One year later, she was diagnosed with complex regional pain syndrome in both legs. She described her pain as severe. She had no other disorders and no family history of chronic pain. She used the black sage sun tea for 1 day and reported that her pain decreased temporarily, then returned. She was later given bilateral lumbar pain shots on four occasions that she said made her back hurt. 

A 28-year-old Caucasian man suffered from two protruding disks (L4–L5–S1) along with a rotated spine and tilted pelvis, which caused debilitating sciatica and muscle spasms. This was the result of two car accidents. He also had chronic pain in his shoulder from an injury several years previously. He had no other medical conditions and no family history of chronic pain. The patient used black sage sun tea for 7 days and physical therapy and has not felt any chronic pain since treatment. The following are the words of the patient: “The first time I used the sun tea, I could feel my pain subsiding comparable to the effect of an anti-inflammatory drug such as an NSAID or steroidal epidural. After soaking my feet in the sun tea, I felt pain relief without the drowsiness side effects that frequently accompany opioids. After the sun tea soak, I felt relieved of my pain and my mind was also clear and not cloudy. I also was able to notice that my shoulder inflammation that I had always had from a prior injury was gone.” In other words, black sage sun tea, along with physical therapy, improved his chronic pain in his back and shoulder. After more than 6 months without pain, the patient reinjured his back at work. His chronic pain returned. He treated himself with sagebrush liniment and black sage sun tea and said it was helpful. He now says he is pain free.

A 62-year-old male Latino suffered from chronic knee pain for 2 years after knee surgery. He had no other medical conditions and no family history of chronic pain. After using the black sage sun tea for 1 week, he reported his pain was gone and did not return. 

A 47-year-old Caucasian female had suffered from plantar fasciitis for 1 month. She had no other medical conditions and no family history of chronic pain. She used the black sage sun tea for 1 week and said her pain was reduced, but still present. Physical therapy was recommended for her to help her learn how to decrease the damage of plantar fasciitis. 

A 55-year-old Latina woman had suffered for many months from plantar fasciitis. She had no other medical conditions and no family history of chronic pain. She used the black sage sun tea for 1 week and said her pain had decreased by 20%.

A 48-year-old Caucasian male had suffered for 2 months from pain due to a damaged rotator cuff. He had no family history of chronic pain and no other medical conditions. He used the sun tea for 1 week and reported that his pain was gone and did not return.

Three arthritis patients used the black sage sun tea for 1 week and reported temporarily decreased pain but no cure of their arthritis. The patients were: a 56-year-old Latina female who suffered for 29 years from arthritis in her leg, a 47-year-old Latino male who suffered for 29 years from arthritis in his hand, and a 54-year-old Caucasian male who suffered for 10 years from arthritis in his hip. None of these patients reported their histories or other medications.

A 49-year-old Caucasian man suffered from a pinched nerve in the neck due to degenerative spinal disease. He had no family history of chronic pain and no other medical conditions. He used the black sage sun tea once and reported that his pain was gone the next day and did not return for about 2 weeks. He later reported that he did not notice any neck pain anymore.

A 68-year-old Caucasian woman suffered from whiplash for several months after a car accident. She had no other medical problems and no family history of chronic pain. She used black sage sun tea for 4 days and reported she was “cured” of her pain. She waited 3 months to report her “cure” since she was certain her pain would return.

A 70-year-old Caucasian woman suffered from chronic neck pain for several months. She did not know what caused her neck pain. She used black sage sun tea for 2 days and reported that her neck was very much improved and continued to be very much improved 2 weeks later.

A 37-year-old Caucasian woman suffered from Morton’s Neuroma. She had no other physical conditions or family history of chronic pain. She used black sage sun tea daily for 6 days and reported that her pain disappeared. She was able to do a week-long backpacking trip with manageable pain. She continued to have no pain several weeks later. 

A 70-year-old Asian man suffered from polymyalgia rheumatica for several weeks and lost more than 20 pounds since he was too tired and in pain to eat. He had no family history of chronic pain and was not using any medications. He tried acupuncture and Chinese herbs and found good, temporary pain relief. He then tried both black sage sun tea for 1 week and sagebrush liniment for several weeks. He said the sagebrush liniment provided good pain relief. After several weeks, he stopped using all pain medications, was no longer in pain and returned to his normal life.

## 4. Discussion

All of the patients reported pain relief after using the *S. mellifera* foot bath. Some of the patients suffering from chronic pain reported long-lasting pain relief with no return of their pain. Arthritis patients reported that their pain returned.

### 4.1. How Does Black Sage Sun Tea Work?

*S. mellifera* contains 54 monoterpenoids, which are 10-carbon compounds [3]. Many of these monoterpenoids stop pain by inhibiting TRP channels [20,21,22]. They quickly penetrate the skin and can inhibit the production of IL-17 and chemokines and down-regulate COX-2 [23,24]. This stops the pain chemokine cycle and improves chronic pain. The tanshinone diterpenoids found in *S. mellifera* are powerful inhibitors of IL-17 expression [25,26]. Inhibition of IL-17 expression stops chemokine production in the skin of the feet. Tanshinones may also directly inhibit chemokine expression [27]. Nerves from the feet communicate with the brain that shuts down chemokine production in the brain and other skin sites. One of the diterpenoids, rosmanol, also inhibits COX-2 expression [28], which decreases prostaglandin release in the skin. This stops the pain chemokine cycle and improves chronic pain (Figure 1).

None of the patients in this study or published studies reported toxicity or side effects from *S. mellifera* sun tea during the week of therapy or at any time after stopping therapy. Allergic dermatitis and other allergies are always a possibility with any plant medicine.

### 4.2. The Placebo Effect

The placebo effect is very useful in pain patients. In fact, topical placebos seem to be superior to oral placebos [29]. Some chronic pain patients are not willing to be healed of their chronic pain [30]. Perhaps their identities are defined by their pain, causing them to develop negative emotional motivational behaviors. These patients may stay on opioids for years, even when their pain increases due to opioid-induced chemokine production which may cause opioid-induced hyperalgesia [17]. In these patients, sagebrush liniment can help with opioid addiction and hyperalgesia. 

Chemokine receptors in the brain may be involved in learned responses [17,31]. Changes in the expression of chemokine receptors could be involved in the placebo and the nocebo effects, which are learned responses. The expression of chemokine receptors is regulated by chemokines and opioids [17]. Chemokines and their receptors are important in opioid-induced hyperalgesia [17]. Chronic pain may be a learned response that involves mesolimbic and prefrontal neurons of the brain [32]. Long after the cause of the pain is gone, the patient continues to feel chronic pain. This chronic pain may be a learned response that may involve chemokine release in the brain and the skin. To date, there is no evidence that placebos decrease chemokine production [33].

Are the effects of black sage sun tea due to the placebo effect? Some of the effects may be due to the placebo effect. The placebo effect is important with any pain medicine. However, many of the patients treated with black sage sun tea were convinced they would get no benefit. Some were adamant about this. The placebo effect has little effect in patients who are sure there can be no benefit. 

### 4.3. The Exercise Effect

Chronic pain patients were advised to perform 20 min of moderate exercise, such as walking, every day. Exercise in combination with *S. mellifera* appeared to help chronic pain in our patients. Exercise alters chemokine levels and chemokine receptor expression in the body [34,35]. Exercise also alters chemokines in the skin and increases IL-15 which is anti-inflammatory [36,37].

### 4.4. Does Black Sage Sun Tea Actually Cure Chronic Pain?

The answer to this question is yes, in some patients, and no in other patients. Patients suffering from back pain due to diabetic neuropathy, also called radiculoplexus neuropathy, suffer from ongoing nerve damage and probably cannot be cured of their pain. Curing chronic pain can be a long process. Since the pain chemokine cycle is a learned process, it must be unlearned. Published work on tension myositis syndrome shows how important it is to unlearn chronic pain [38]. Chronic pain can involve fear, anger and other mind–body concerns that, in the experience of the authors, must be alleviated in order to provide a long-term treatment of chronic pain. The patient must learn how to unlearn chronic pain and be aware of how easily chronic pain can return. Unlearning chronic pain is just as important, in many patients, as using black sage sun tea to relieve pain.

## Figures and Tables

**Figure 1 medicines-06-00018-f001:**
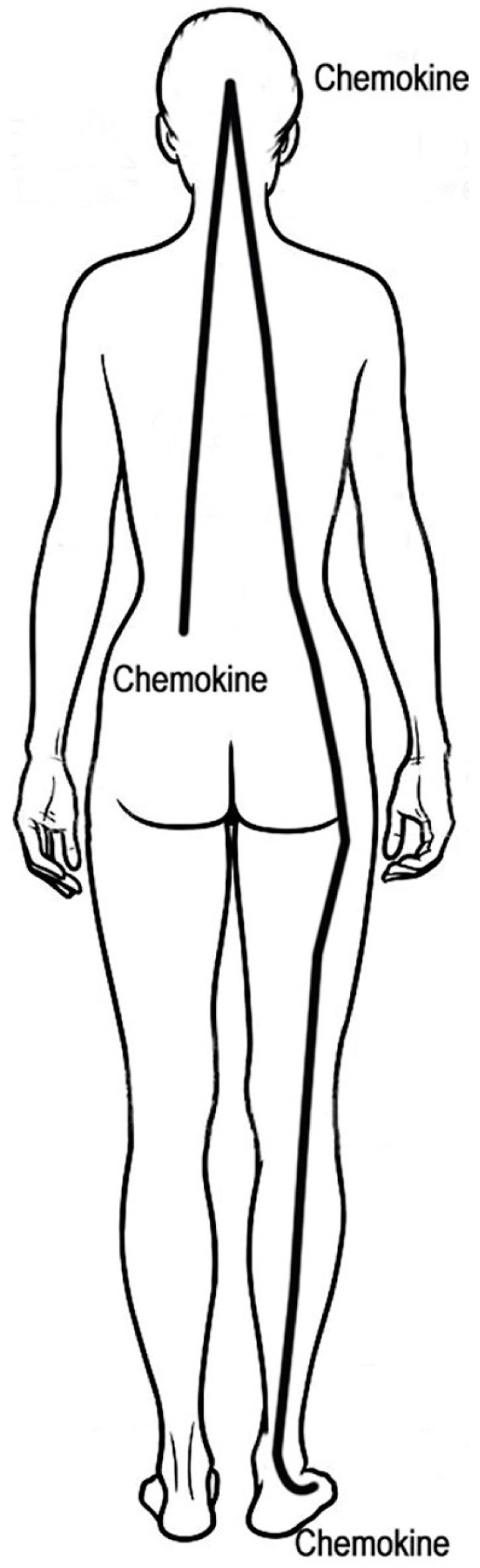
Proposed pain chemokine cycle from the skin to the brain and returning to the skin. The initial pain causes chemokine release in the skin, such as the lower back, which attracts monocytes, neutrophils and other inflammatory cells to the skin. Chemokines increase pain by lowering the action potential threshold of sensory neurons. Chemokines induce prostaglandin and IL-17 release in the skin, both of which increase chemokine release. Skin sensory neurons may stimulate the release of chemokines in the brain. Brain chemokines may modulate descending and peripheral neurons, resulting in chemokine release in the skin at the site of initial pain and other sites, such as the feet. This suggests that treatment of the feet may decrease chemokine production in the back, brain and other sites. There are many other macrophage- and neutrophil-derived cytokines involved in pain and inflammation such as tumor necrosis factor, IL-1β, IL-3, IL-5 and IL-6 [18]. These cytokines may increase prostaglandin, norepinephrine and leukotriene B_4_ release in the skin, which enhances and prolongs pain [9,10]. Leukotrienes activate TRP channels to cause and prolong pain [19].

**Table 1 medicines-06-00018-t001:** Patient data summary. All patients reported initial pain of 6–10. Most had decreased pain of 0 or 1 after treatment. DTS indicates declined to state. Partial indicates partial relief of pain of 2–4.

Age	Site of Pain	Family History of Pain	Duration of Pain	Duration of Pain Relief
20	Legs	No	A few months	1 day
28	Back	No	Several months	Several months
62	Knee	No	2 years	Several months
47	Foot	No	1 month	Partial
55	Foot	No	Several months	Partial
48	Shoulder	No	2 months	Several months
56	Leg arthritis	DTS	Several years	1 day
47	Hand arthritis	DTS	Several years	1 day
54	Hip arthritis	DTS	Several years	1 day
49	Neck	No	Several years	Several months
68	Neck	No	Several months	Several months
70	Neck	DTS	Several months	Several weeks
37	Foot	No	Several years	Several months
70	Torso	No	Several weeks	Several months

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
