# Peer review of "Salvia mellifera—How Does It Alleviate Chronic Pain?"

_medicines, 2019, doi:10.3390/medicines6010018_

Round 1
Reviewer 1 Report
This is the 3rd version of this manuscript. The authors make improvements each time. This version did not contain a response to review comments on the previous version, but it is encouraging to see that many of the issues raised have received attention. There are still details that need attending to - some are minor, but some are still substantive.
Line 29: pain (6,7).
Line 40: “Sagebrush liniment is more powerful than morphine” – what is the evidence for this statement? It does not need to be included, and can simply be deleted.
Line 48: PGs also cause phosphorylation of Na+ channels, and the relative involvement of TRP and Na+ channels has not been determined. Please modify phrasing.
Line 58: “ … including descending neurons, which results in chemokine release in the skin”. Descending pathways go to the spinal cord. Peripheral output is from the spinal cord. Please be less certain about how chemokines might be involved.
Figure 1 remains speculative. If you must include some version of it, please Make the speculative nature of the proposal clear in the figure legend.
Section 3, regarding nature of treatments. Once one gets to the Discussion (4.1.3), there is mention that “patients were advised to perfor4m 20 min of moderate exercise, such as walking, every day.” This is the first mention of an involvement of exercise in the current report. It needs to be in the description of the treatments in Section 3.
Line 113: Start a new paragraph for the 55 year old, to continue as for others that came before.
Lines 120-124: This paragraph describes 3 subjects but only includes who they are. Please comment on their outcomes, as for all other cases.
Line 146. Discussion. Before getting to the mechanism, make a brief summary of case outcomes reported in the body of the paper.
Line 151: Link line 152 here so that the section is only one paragraph.
Line 158: Delete the statement “Placebos are the safest drugs.” Given that nocebo effects can occur, this is not accurate. It is an opinion.
Lines 159-161: “Many patients…. Placebo effect.” Delete these lines – the content is not relevant to the current report.
Line 172: “This learned response probably involves….” This is speculative. The author seems adamant about the mechanism involved, but really, there is nothing to directly support this. It is purely speculation. Wording needs to be cautious.
Lines 179-188: Delete this section. The issue of expectancy, a factor involved in placebo, appears at the end of 4.1.1.
Line 195: “is yes in many patients”. Hardly, the current report is of 14 patients out of “probably between 100 and 200” (Line 84). More accurate phrasing is “some patients”.
Section 4.2 can be deleted. This is speculative.
Author Response
Line 29 - has been changed as requested
Line 40 - I have added - contains cineole that is more powerful than morphine (11)
Line 48 - I have added - Phosphorylation of Na+ channels is also induced by prostaglandins (15) which may make them more sensitive to stimuli. Both TRP and Na+ channels, as well as many other skin receptors, are important in pain (10).
Line 58 has been modified - Chemokines in the brain modulate the actions of other neurons including descending and peripheral neurons, which may result in chemokine release in the skin (17).
Figure 1 - The purpose of this manuscript is to stimulate discussion of possible new mechanisms that may lead to new ways to treat pain and chronic pain. The figure 1 legend has been modified. Line 66 has been modified - The brain may be involved in the pain chemokine cycle (Figure 1).
Section 3 has been modified - Patients were also advised to perform 20 min or so of mild exercise daily, such as walking.
Line 113 is now a new paragraph.
Line 120-4 has been modified - reported temporarily decreased pain but no cure of their arthritis.
The discussion has been modified with a new lead paragraph - All of the patients reported pain relief after using the S. mellifera foot bath. Some of the patients suffering from chronic pain reported long lasting pain relief with no return of their pain. Arthritis patients reported their pain returned.
Line 151 and 2 are now joined into one paragraph.
Line 158 has been deleted.
Lines 159-61 have been deleted.
Line 172 presents a possible new mechanism in chronic pain that may lead to new treatments for chronic pain. Line 172 has been modified - This chronic pain may be a learned response that may involve chemokine release in the brain and the skin.
Lines 179-88 have been deleted.
Line 195 has been changed as suggested.
Section 4.2 - the nocebo section has been deleted.
Reviewer 2 Report
the presented data have no scientific support
there is no proper description of the scale use to evaluate pain
Feet bath is improbabile to cure chronic pain with undocumented medical history of the patients
there is no conclusion section
i would suggest a non scientific journal for publishing this paper
Author Response
I give the mechanisms/scientific support involved in this pain medicine. The medicine is a traditional medicine that has been used for thousands of years by Chumash Indians.
The 0-10 pain scale was used as normal.
I give as much information as possible about the medical history of each patient. I give evidence to show that a foot bath can provide long term relief of chronic pain.
There is a discussion section. The first paragraph contains the conclusions.
This is scientific work.
Reviewer 3 Report
An excellent contribution to the literature.
However, it will need further work before being ready for publication.
Introductory paragraph on pain is too simplistic. Recommend collaboration with a pain fellow
at UCLA or other such professional engaged on a daily basis in pain management may perhaps be helpful. Specifics follow:
Introduction:
1. "Chronic pain" paragraph needs to be rewritten. Chronic pain is not one entity but encompasses a myriad of conditions. Likewise, "opioid therapy" is not the same as the "opioid crisis" which is largely socially driven,so this needs to addressed with more finesse. eg. Opioid therapies are commonly utilized in the management of many painful conditions, including osteoarthritis, back pain, oncological pain and others.....
The use of adjuvant therapies such as salvia mellifera, may form an important adjuvant to consider in the management of painful arthritis, or musculoskeletal pain.
2. Topical medicines are not harmless either. Avoid blanket statements that you are not specifically proving, or explicating.
3. Distinguishing and explaining the difference between sagebrush liniment and the sun tea, and how each are prepared, or where they can be obtained, as well as the dosage, perhaps in a "Method of preparation" section. Also for how long can it be used? Any long term effect with long-term use? Allergic potential?
4. Cytokine figure - needs much more detail. Refer to pain management texts or other articles discussing pain and cytokines for information. Can also make your own diagram based on the molecules mentioned in the discussion.
5. A discussion of any side effects to watch out for, can it be overdosed, how pure are the commercial sources, etc.
Discussion and conclusion: Some change in tone is needed eg. Sage liniment and sun tea may be important adjuvants to consider in the management of painful conditions such as arthritis, musculoskeletal pain or oncological pain (or something else?)
Author Response
I cannot collaborate with a pain fellow. This work is not FDA approved.
The chronic pain paragraph has been rewritten.
I have included statements about potential hazards of topical therapies.
I have put in a brief description of the published work on sagebrush liniment in the methods section as requested. None of these preparations are commercially available. The length of use and methods of use are described in the methods section or references contained in the section. Allergies to plant medicines are always possible. This is now included in the discussion.
I have changed the figure legend. I have added some discussion of other cytokines in pain.
I have added a discussion of side effects. There are no commercially available products.
I have changed the tone of the discussion. Although the sun tea is anti-inflammatory, I do not believe it is an immune stimulator and probably has no use as an adjuvant.
Round 2
Reviewer 3 Report
Would delete the figure, as does not offer additional information, and too simplistic.
If wish to have a figure, it will need to be significantly revised, however I don't believe it
is needed, as does not offer additional information that is not in the text.
Rest of paper is OK to proceed.
Author Response
I have added to the figure legend to point out the importance of leukotriene activation of TRP channels in chronic pain.